# Measuring Characteristics of Explanations with Element Maps

**Steffen Wagner** * , **Karel Kok** and **Burkhard Priemer**

Department of Physics, Humboldt-Universität zu Berlin, Newtonstraße 15, 12489 Berlin, Germany;
karel.kok@physik.hu-berlin.de (K.K.); priemer@physik.hu-berlin.de (B.P.)
* Correspondence: steffen.wagner@physik.hu-berlin.de

**Abstract:** What are the structural characteristics of written scientific explanations that make them good? This is often difficult to measure. One approach to describing and analyzing structures is to employ network theory. With this research, we aim to describe the elementary structure of written explanations, their qualities, and the differences between those made by experts and students. We do this by converting written explanations into networks called element maps and measure their characteristics: size, the ratio of diameter to size, and betweenness centrality. Our results indicate that experts give longer explanations with more intertwinement, organized around a few central key elements. Students' explanations vary widely in size, are less intertwined, and often lack a focus around key elements. We have successfully identified and quantified the characteristics that can be a starting point for guiding students towards generating expert-like written explanations.

**Keywords:** element maps; expertise; explanations; networks; optics phenomena

## 1. Introduction

Explaining natural phenomena is a core practice for scientists [1–4], and thus provides an opportunity for designing learning activities that reflect scientific practice [5,6]. Accordingly, it is incorporated into several curricula and policy documents [7–10]). Explanations constitute a fundamental part of scientific knowledge [11] in which observations and theoretical concepts are tied together [12]. Connecting aspects of observations with theory is an endeavor for both scientists and science learners [13–15]. Especially science learners struggle with this [16–20]. To guide students towards better explanations, it would be useful to have more tools that give insight into the structure and quality of explanations.

According to Rocksén [21], explanations appear in three different contexts: scientific explanations as discourse or linguistic products (e.g., [11,22]), explanations as instructional practice in science lessons (e.g., [23,24]), and explanations within an everyday context (e.g., explaining why I prefer blue shirts to red). For science education, the first two approaches are relevant. Since in our study, we focus on university students' written explanations of natural phenomena, we will make use of the first—linguistic—perspective. From these perspectives, the written explanations can be viewed from different standpoints, e.g., from the argumentation view [25] or from the model view [26]. However, in each of these standpoints, explanations have to be communicated. They have a structure and thus can be analyzed from a linguistic perspective.

Research on the structure of explanations has thus far focused predominantly on their internal coherence [27], the use of specific elements [28], the interaction between different representational modes [29], or the use of evidence [30]. However, as explanations connect aspects or elements of observations with those of theories, how does this connection work in detail? How are all the single entities and relations between these aspects or elements represented in an explanation? Little is known

about this elementary structure in experts' explanations, nor is it clear how it differs from those given by students. Gaining insight into the elementary structure would allow for an analytical comparison of the quality of experts' and students' written explanations from which students could benefit [31].

Networks are structure representations by definition [32]. As graph-orientated approaches, they have become a valuable tool for both describing expertise and revealing scientific knowledge structures for educational purposes [33–35]. In these contexts, networks always have a graphical structure and contents, each with an individual level of quality [36]. For a complete analysis, both these qualities would have to be considered.

In this paper, we will visualize the entire elementary structure of written explanations in so-called *element maps* (our element maps should not be confused with those from Clark [37], which are tables rather than networks, and are built for other purposes). We will focus on the quantifiable characteristics of these element maps, and what this means for the elementary structure of the written explanation. In the next two chapters, we report in detail on how an element map is created and how it differs, for example, from concept maps. We describe the graphical structure of element maps using four different network characteristics. With these characteristics, we interpret the features of experts' and students' written explanations and compare them. The context for this will be the common optical phenomenon of apparent depth.

## 2. Theory

Explanations are discourse products. They consist of parts made up of different representational modes [38], e.g., written text, images, and formulas, where the main part consists of written text. A written text is composed of terms representing abstract entities, i.e., thoughts or ideas, that stand for real or imaginary objects, systems, and properties [39]. It also contains descriptions of how these entities are related to each other, forming scientific statements. We call this the *elementary structure* of a written explanation.

In a written text, an entity can be represented by different words (synonyms), and the same word could represent different entities (a homonym). Additionally, these words could appear throughout the text. This forms a problem for having access to the elementary structure of a written explanation because, in the element maps, every entity will only appear once. It is also hard to identify the key ideas or key relations as well as their concatenation in the text easily. Networks are an appropriate approach to solve these problems.

A network consists of vertices (or nodes) and edges, where every starting and endpoint of an edge is a vertex. An established application of networks in science education is concept maps. In concept maps for scientific knowledge, the labeled vertices represent scientific concepts and the labeled edges represent the relations between them [34]. Here, concepts are more or less complex constructs that, for their part, could be sub-structured in entities and relations too. For example, the ray-concept of light contains light rays, directions, angles, etc. in different possible constellations.

By contrast, we do not aim to visualize the conceptual structure, but the written explanation's elementary structure of all entities and relations—which together make up the single elements. This results in a major difference compared to concept maps: in element maps, both entities and relationships are shown as vertices. The edges in an element map represent which vertices are connected. Hence, element maps have a higher resolution compared to concept maps. This is due to the structure of written language. A vertex can be both the source or target of an edge. Source, target, and edge together form a proposition. In language, a proposition can again be the target or source of another relationship and form a higher-level proposition. While a vertex can belong to several propositions, a relationship can be uniquely assigned to the respective proposition. Thus, the relationships are the only source and target points that can connect to higher-level propositions and therefore directly fulfill the definition of a vertex. We will illustrate this with an example in the following section.

Another key difference with concept maps is the origin of the map. Concept maps are generally created by students and experts themselves, explicating their conceptual understanding. With element

maps, the participants do not create these maps themselves, but write an explanation from which we build maps.

The difference between concept maps and element maps is summarized in Figure 1.

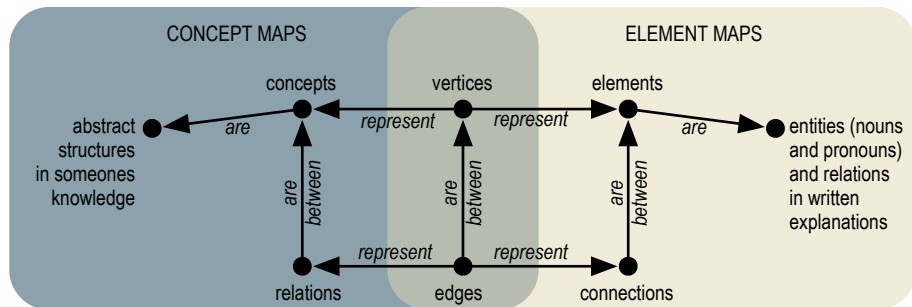

**Figure 1.** Comparison between concept maps and element maps. Concept maps are created by individuals from their conceptual understanding, whereas element maps are generated from individuals' written explanations.

Our way of extracting the elementary structure of a written explanation and visualizing it in an element map will be described briefly in the Materials and Methods section. However, we can make use of the valuable experiences from the great body of concept map research, for example, on ways to interpret network characteristics.

Once the elementary structure is visualized in an element map, the following established network characteristics can be analyzed: (a) the size, b) the diameter, (c) the ratio of diameter to size (intertwinement), and (d) betweenness centrality. Here, we briefly describe these measures.

(a)  The size, $s$, of a network is the number of its vertices.
(b)  The distance between two vertices in a network is the smallest number of edges that connect these vertices. The diameter, $d$, is the largest distance in a network.
(c)  There are different measures for the level of intertwinement (e.g., density and average path length) for different purposes. By intertwinement, we mean the level of complexity, or how interwoven the network is. We use the ratio of the diameter and the size, $d/s$, as our measure for intertwinement.
(d)  A variety of established centrality measures for vertices quantify how important a vertex is for a network from different perspectives. We use the betweenness centrality, which indicates how many distances a vertex is placed [40]. Vertices with a high betweenness centrality act as mediators that tie together different parts (e.g., those representing theory and the phenomenon) of the network and form central elements around which the network is formed.

These and other features are established characteristics, applied in concept maps or other network visualizations in science education, too [33,35,41–43]. Concept maps created by experts differ in many aspects from those made by students. For example, experts build larger maps to represent their conceptual understanding [44,45], their concepts have more interrelations [44], and experts make more use of key concepts [34,42,46]. For an overview of the use of network characteristics in science education, see, e.g., [36,41].

Concept maps have proven to be helpful tools in science education. Structural characteristics of concept maps are indicators for the quality of conceptual understanding. So far it is unclear if the same holds for element maps and the quality of written explanations. To analyze this, we start by constructing element maps from written explanations and measure their structural characteristics quantitatively. Element maps of experts' and students' written explanations are then compared.

*Research Question*

Accordingly, our research questions are as follows:

1. Which structural characteristics do explanations of a phenomenon given by experts and students have from the perspective of the network approach in element maps?
2. What differences emerge in the networks between experts and students?

## 3. Materials and Methods

We collected 19 written explanations from students (university freshmen in physics and biophysics), as well as 3 experts (science education researchers). We chose to select science education researchers rather than physicists as experts because the apparent depth is an everyday phenomenon and not a phenomenon that is the subject of current physical research. Additionally, the experts involved have dealt with this context in an educational context as well as in publications.

The explanations were written and analyzed in German. The context of the explanation is the phenomenon of apparent depth (see Figure 2). This everyday phenomenon makes objects appear closer to the water surface than they are when observed from outside of the water. This phenomenon is often explained using Snell's Law and the refraction of light that takes place when light travels from one medium to another, in this case at the interface of water and air [47]. One way of explaining this phenomenon is by describing the light traveling from the coin to the observer's eye, bending at the interface of water and air. For the observer, this results in an upward shift of the image of the coin as compared to where the image would be without the water.

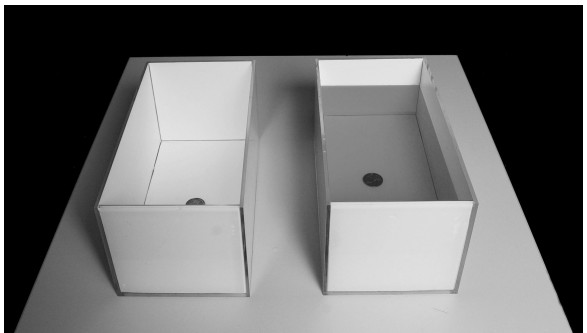

**Figure 2.** The phenomenon of apparent depth.

The explanations were written on a laptop. Sketches or other drawings could be made using paper and pencil. These were used as support for the analysis of the maps' content and in the case of an unclear meaning of terms, and were not separately analyzed. This project is part of a larger survey, in which a total of 64 explanations for different phenomena were gathered (see [48]). The identification of the elements was done by two raters with a relative inter-rater agreement of $r = 0.83$. Since this is an identification and not a categorization of elements, the specification of a measure for inter-rater agreement considering the possibility of agreement occurring by chance (e.g., Cohen's Kappa) can be dispensed in this case.

*Generating Element Maps*

Next, we will show an example of how a sentence from an explanation of the apparent depth phenomena can be structured and represented in the form of an element map. The sentence that we will use is as follows:

"The coin on the right seems to be lifted for the observer because the light is refracted at the water surface."

This example sentence contains the following entities (nouns and pronouns): "coin (right)," "observer," "light," and "water surface." Synonymously used words will be included in the same entity.

Between the entities "coin (right)" and "observer" is the relation "seems to be lifted," and between "[the] light" and "water surface" is the relation "is refracted at." This gives rise to the two

propositions "coin (right) seems to be lifted for [the] observer" and "[the] light is refracted at [the] water surface." These propositions are then connected by the relation "because," forming a new proposition. The word "because" can—like every relation—appear multiple times in an element map because it can be used in numerous causal connections. Nevertheless, there will be only one element for every distinct entity or relation in an element map. The multitude of connections will indicate the repeated use of this element. The consolidation of entities with the same meaning implies that the repetition of the same statement does not lead to a larger number of vertices.

To form the network, we first place all the entities as vertices in the map (gray) (see Figure 3). The relations between these entities are depicted below them as vertices (orange), the relations between the propositions are shown below that, and so on. This way, all the elements—i.e. all entities and relations—are represented as vertices in the network.

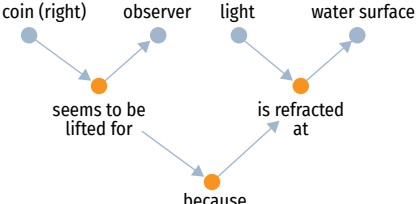

**Figure 3.** Element map of the example sentence. Entities and relations are indicated by vertices with different colors.

The next step is to connect all the elements with so-called edges. A clear difference between concept maps and element maps is that, in concept maps, relations (such as "seems to be lifted for") are depicted as labeled edges, whereas in element maps they form their own vertices. The reason to do so is that the relations can be the source or target of subsequent relations (see Figure 3), making them vertices by definition. However, if one adds a sentence re-using the entity "observer," the existing vertex will become the source or target vertex for another proposition, and so on. In this way, the element map grows as sentences are added until the entire explanation is fully captured.

Because we focus on quantifiable characteristics of the maps and not on the content of the explanations, we do not take into account an analysis of the propositions' correctness and the presence of non-relevant elements. With this limitation, we do not unfold the full potential of element maps. Nevertheless, this structural-only approach can give valuable insights that can be supplemented by a content analysis of the individual explanations. For a full description of a complete approach including content analysis and expert validation, see [48].

As our focus lies on the structure, we will express this structure using several quantities that will be explained in the following subsections. These quantities, however, do not require the level of detail that is shown in Figure 3. Therefore, we can simplify the representation of the maps. This is done by removing the labels and colors of the vertices as well as the direction of all the edges in the graphical representation (see Figure 4). Moreover, the placement of vertices will, in general, not have a hierarchical structure such as Figures 3 and 4. This information is, however, still available permitting the possibility of a content-based analysis.

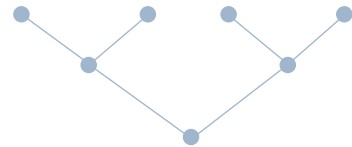

**Figure 4.** Element map of the example sentence without labels and colors.

## 4. Results

The next subsections will show the results of map analysis for students' and experts' explanations. The results of the size, diameter, and their ratio are depicted as boxplots in Figure 5. Because of the limited amount of data on expert maps, we will not look into the significance and effect size of the differences.

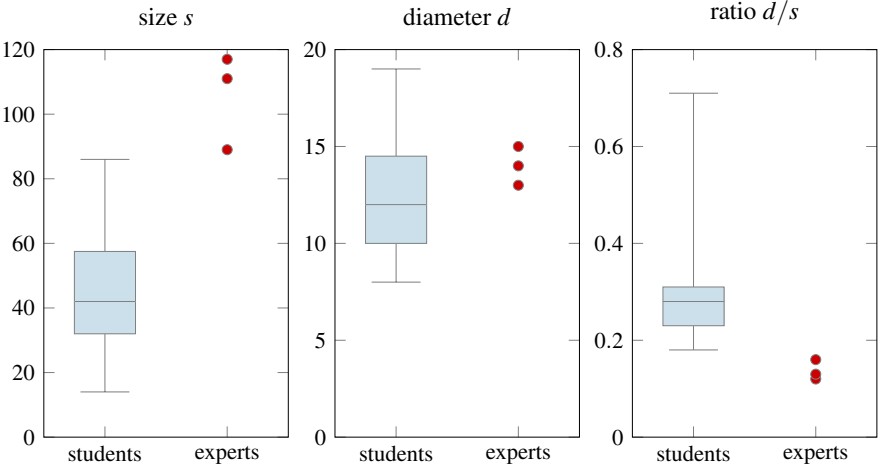

**Figure 5.** Formal network characteristics visualizations for students' and experts' explanations. Whiskers indicate the extremes.

### 4.1. Size

Figures 6 and 7 show typical element maps of a student and an expert explanation. The student map has a smaller size as compared to the expert map. This holds for our entire sample, as depicted in Figure 5 (left). The size of student maps varies much more strongly than that of expert maps.

Additional analysis of the maps shows that the ratio of the length of a text (the total word count) to the size varies between 2.2 and 3.6. For example, we have an explanation with a size of $s = 41$ with a length of 92 words and another explanation with a size of $s = 42$ with a length of 151 words. This means that the length of an explanation is not a direct measure of the size of the network, which can vary by a factor of 1.6.

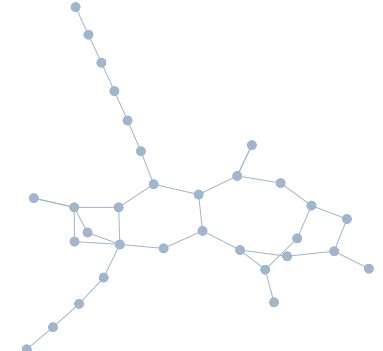

**Figure 6.** Student map with size $s = 32$.

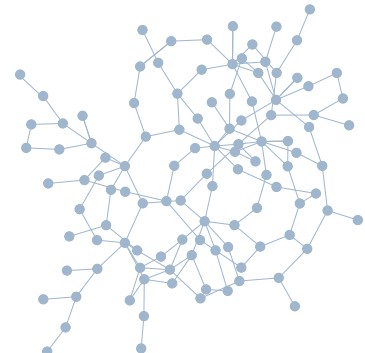

**Figure 7.** Expert map with size $s = 111$.

### 4.2. Diameter

Figures 8 and 9 show the visualized diameter of the same student and expert map as shown in Figures 6 and 7. Although the expert maps fall in the same diameter range as the student maps, the expert maps have a much smaller variance, which can also be seen in Figure 5 (middle).

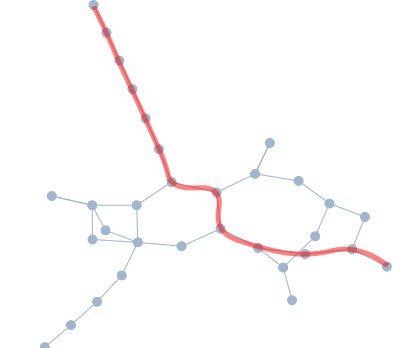

**Figure 8.** Student map with diameter $d = 12$.

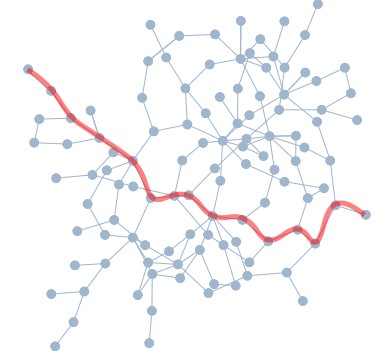

**Figure 9.** Expert map with diameter $d = 13$.

### 4.3. Ratio Diameter/Size

As can be seen in Figure 5 (right), expert maps have a lower ratio of diameter to size. This means that they look much more compact than student maps, or are intertwined much more strongly. Student maps, however, most often do not do this, resulting in a less intertwined, looser element map.

### 4.4. Betweenness Centrality

In Figures 10 and 11, the betweenness centrality for the two typical maps is represented by the size of the vertices. It can be seen that the expert map has a few large vertices—i.e. with a large betweenness

centrality—that represent key elements that form a central role in the map. On the other hand, the student map is made up of small vertices that do not vary in size much.

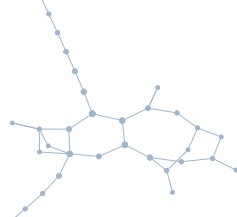

**Figure 10.** Student map with betweenness visualized as vertex size.

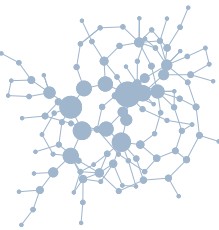

**Figure 11.** Expert map with betweenness visualized as vertex size.

This is confirmed by Figure 12, in which boxplots for the betweenness centrality of all vertices are shown for all our student and expert maps. What is clearly seen is that all expert maps have positive outliers, whereas there are only a few and less extreme outliers for the student maps. These outliers represent key elements with an exceptionally high betweenness centrality.

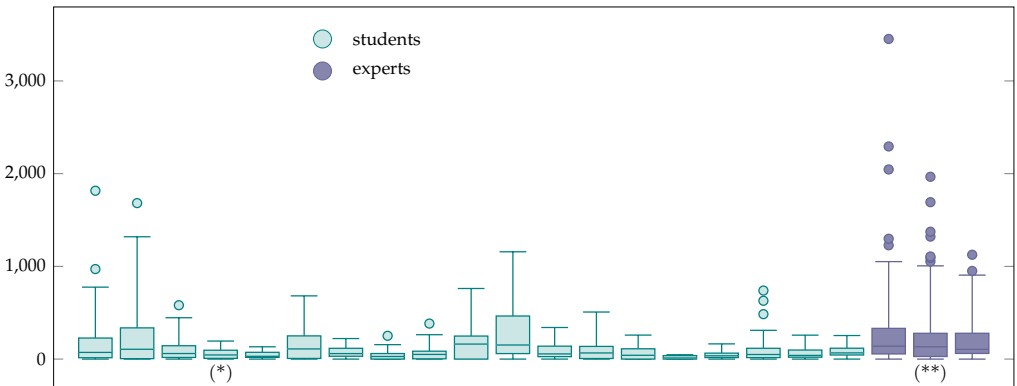

**Figure 12.** Boxplots for betweenness centrality of vertices for student and expert maps with whiskers indicating $3 \cdot IQR$ (interquartile range). Outliers indicate vertices that fulfill a central role in tying together a written explanation. The network marked with (*) is the student map, and the one marked with (**) is the expert map, as shown in the figures above.

## 5. Discussion and Conclusions

In the next subsections, we will discuss the results of the analysis of the network maps. We will start with the interpretation of relevant results and compare them to existing research. After that, we will talk about the implications and how we can help students in writing better explanations.

*5.1. Size*

The large size of expert maps as compared to student maps seems plausible to us. Larger maps represent more content knowledge as well as a more thorough understanding of the phenomena,

both of which are expert characteristics [49,50]. As such, the size of an element map can be seen as a measure for the amount of knowledge elements integrated into the explanation. It should be noted that another quality measure of written explanations by experts is their parsimony [12]. This indicates the need for future content analysis to measure the relevance of all the elements, as well as their correctness, appropriateness, and redundancy.

We explain the high variance in size measure for student maps by the differences in content knowledge and differences in the ability to apply this knowledge to a written explanation of a phenomenon. In the context of concept maps in science education, a greater conceptual understanding is associated with larger concept maps [41,45]. The size of element maps allows us to measure a similar quantity—the number of knowledge elements integrated into a written explanation—by means of the size.

The size characteristic of an element map can be used to give feedback on students' written explanations. When students write explanations with a small network size, they could be advised to try and include more relevant theoretical concepts (e.g., refraction, light rays, and images), more descriptions of the phenomena (e.g., placement of the coin, place of the observer, the coin's image), or more relations between them in their written explanations.

As our additional analysis has shown, the ratio between the size characteristic and the actual length of a written explanation varies strongly. This holds for students as well as experts. What we can conclude is that the length of a text does not necessarily correspond to the size of the explanation. With element maps, however, we can reveal this size in a more precise manner than just by looking at the length of the written text.

## 5.2. Ratio Diameter/Size

The ratio of the diameter to size appears to be a more relevant and meaningful characteristic for describing the elementary structure than the diameter alone. The small ratio for expert maps in comparison to student maps seems plausible to us. Growth in structural knowledge is associated with more cross-connections between individual concepts [51]. This trend is also found in the analysis of concept maps [41]. When transferred to the written explanations, the smaller ratio for experts indicates more intertwined explanations with more cross-connections between nouns and propositions, presumably resulting from very dense knowledge structures, whereas the larger ratio for students indicates fewer cross-connections, presumably resulting from less dense knowledge structures. Of course, such statements can only be confirmed after a content-based analysis. A quick look into the content analysis revealed that students in this sample predominantly used correct propositions and relevant entities. No errors were found in the three expert statements.

One way to help students in writing a more intertwined explanation might be to have them think about different relations between entities or, in other words, build and write down propositions before writing down the explanation. This could be achieved with a scaffolding of several directed questions towards building propositions. For instance, how do the object and the image relate? Or, how can you describe the path of the light ray using the law of refraction? This scaffolding does not necessarily have to be phenomenon- or context-dependent. A scaffolding for a stepwise structuring of a written explanation can also be successful at helping students write explanations [52].

## 5.3. Betweenness Centrality

In the distribution of the betweenness centrality of vertices in expert maps, we see a small number of vertices with an exceptionally high betweenness centrality, whereas the rest is comparable to those in student maps. Since explanations connect observations or phenomena with theory, it is reasonable to assume that elements with a high betweenness centrality act as mediators between the phenomenon and the theory. This is a hypothesis that could be tested by classifying the elements according to their assignment to the phenomenon or theory (see, e.g., [48]). Possible outcomes would provide further (and measurable) information about a critical point in students' conceptual understanding since we know that the connection between theory and observation is often an obstacle for students. However,

the high value of the betweenness centrality of some vertices suggests that experts structure their explanations around key elements. In contrast, students' explanations lack a focus on central elements.

From a content analysis, we can also see that different experts will make use of different key ideas. When looking at the contents of the central vertices of our three expert maps, we find that for one expert map the two most central elements are "observer" and "light." For another expert map, these are "eyes" and "light rays," and for the last one these are "coin" and "interface."

In the research on concept maps, a similar effect is seen with key concepts [34,46]. Except when exposing the elementary structure of an explanation, it is not the concepts, but the entities and their relations (the elements) that are important. Moreover, in concept map research, a hierarchical structure is often assumed. In element maps, hierarchy is an unimportant arbitrary choice of layout. Betweenness centrality emerges as a measurable property of the written explanation. Therefore, the usage of betweenness centrality seems appropriate for describing the quality of a written explanation. The visual and quantifiable representation of key elements is a unique feature of these element maps.

The presence of central elements highlights the relevance of using key ideas (see, [9,53]), and adds the notion that entities, as well as their relations (which together might later construct a concept), can fulfill this role. There are several ideas to guide students towards working with key elements in science teaching [54]. For an application in a learning situation, this requires the identification and selection of key elements by the educator or in a guided group discussion beforehand. These key elements could be the starting point for the directed questions in the scaffolding mentioned above.

## 6. Limitations and Outlook

Our findings open up two perspectives, one for teaching and learning and one for further research. By looking at the different characteristics of the element maps, teachers can provide initial guidance as to whether elements are lacking in the explanations, whether more cross-linking is necessary, or whether the emphasis on key ideas is missing. Which entities, relationships, and key ideas these are can only be determined in connection with an analysis of the content. For learning purposes, educators should also include a content analysis of the element maps, e.g., pointing out incorrect propositions and irrelevant elements. Another possibility for learning how to structure a good explanation is to use element maps as a starting point for generating a written explanation. For example, one could show students an expert map and have them figure out what the key elements are (i.e. those with a high betweenness centrality) and how they are related to each other. The identification of key elements, their relations, and their intertwinement could be implemented into a scaffolding environment for improving explanation skills, similar to [52].

There are several frameworks that structure explanations. Our element maps are independent of these frameworks and offer the opportunity of revealing the underlying elementary structure of explanations within these frameworks. For example, the CER framework [25] structures argumentations and explanations into claims, evidence, and reasons. Each of these instances can consist of several statements, and each statement is again characterized by relations between objects, properties, and changes. These all consist of entities and relations, the very elements that our element maps can visualize and measure. From the model perspective of Passmore [26], explanatory models connect aspects of phenomena with those of theory. With a respective categorization of the elements in element maps, it would be possible to assign single elements to phenomena and theories and locate students' difficulties in (or between) the two domains (see, e.g., [48]).

The research presented here is an exploratory first step in visualizing and analyzing the elementary structure of written explanations. From our results, we can state new hypotheses. For example, the quality of a written explanation can be assessed using (a combination of) the network measures size, betweenness centrality, and the ratio of size to diameter—the higher the values, the higher the quality of the written explanation. This, in turn, means that there are strong analogies to the research on concept maps. Furthermore, it is now possible to investigate the development of the elementary structure during the learning process by looking at explanations before and after learning sequences

and how their characteristics change. Furthermore, the results presented here do not include the results of the content analysis. This content analysis, including the correctness of the propositions and the examination of the structural measures, constitutes the basis of the entire procedure. Without content analysis, the analysis of structural features ultimately remains incomplete. Nevertheless, we have identified comprehensive and well interpretable differences between experts' and student' written explanations using these quantitative characteristics.

So far, we have successfully applied element maps to identify and quantify features of explanations that otherwise are difficult to measure. Due to our sample size, the focus on one phenomenon, and by looking at the structural features only here, our results are limited in terms of generalizability. A first step in improving this would be to analyze more expert explanations to quantify the differences between experts' and students' explanations.

The process of creating network maps from written explanations up to 200 words can be done manually in a reasonable amount of time. The analysis of longer and more explanations would require an automated approach. Some promising first steps in this automation process have been taken by the authors in [55], and such steps could be suitable for the special structure needed for our element maps.

Our novel representation of written explanations provides unique and quantitative measures to the quality of their elementary structure. This makes key features of a written explanation accessible and comparable in a numerical, as well as visual, manner. Additionally, element maps provide an alternative approach to addressing abstract features that otherwise would remain hidden in written explanations.

**Author Contributions:** S.W.: investigation, methodology, conceptualization and writing; K.K.: review, writing and editing; B.P.: supervision. All authors have read and agreed to the published version of the manuscript. All authors have read and agreed to the published version of the manuscript.

**Funding:** This research received no external funding.

**Conflicts of Interest:** The authors declare that there is no conflict of interest.

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
