# Peer review of "Measuring Characteristics of Explanations with Element Maps"

_education, doi:10.3390/educsci10020036_

Round 1

Reviewer 1 Report

The paper introduces a network analytical procedure to quantify the quality of student explanations. The authors find network measures that allow to distinguish the explanations from experts and novices and discuss implications for research and practice.

Generally, the paper is well written and the analysis seems sound and the results solid. 

However, when reading the introduction and discussion I felt that key references where missing relative to which the authors should situate their work to make a better contribution. More specifically, I miss a discussion of the influential work of Toulmin on explanations and argumentation in general. In science education, the authors do not discuss the work of e.g. Kulgemeyer or McNeil who introduces a framework for teaching explanations, or Yao who proposes a learning progression for explanations. Similarly, explanation maps by Clark are not mentioned. Since the paper (at least implicitly) makes the argument that the element maps provide new understanding I would like to see a discussion of what it is that we gain (as science education researchers AND for classroom practice) when we use element maps instead of e.g., the CER Framework (McNeil) to judge the quality of an explanation. Such a discussion would largely improve the papers chance of meaningfully contributing to the literature. Right now, a statement like "Our novel representation of written explanations provides unique and quantitative measures to the quality of their elementary structure. This makes key features of a written explanation accessible and comparable in a numerical, as well as visual, manner." does not convince me until I read an argument why e.g. the CER framework does not give me all the information about the quality of an explanation that I would want. 

With this being my main message for the authors and criticism I have a number of minor points here:

I wonder whether science education researchers are really good "experts" for scientific explanations? Why not ask physicists etc? 

There should be a more elaborate in justifying their network measures in the methods section, e.g., the authors say that high betweenness centrality indicates nodes that connect different parts of the networks but do not say why connecting different parts of the element maps may be important in the context of explanations. 

Why focus on the explanations of the apparent depth phenomenon only if data on others was collected?

The authors state (L 145) that they do not judge the normative correctness of the propositions but refer to a dissertation. This is ok, but they should discuss the potential and limitations of doing so more elaborately.

Figure 5 uses box plots to represent the data from 3 experts. as box plots try to depict a distribution this is not warranted here. Instead, the authors should just plot the three data points.

In the discussion (L 223) the authors discuss how their network measures relate to structural knowledge. Basically, this is inconsistent with not judging the correctness of the propositions (L 145). How can we talk about knowledge if the maps, strictly following the methods section and not judging the correctness of propositions, could have been created from totally wrong explanations that are not even on the topic? I understand that the authors do want to focus on the structural aspects but somehow this inconsistency needs to be solved, as in the current state the interpretations in the discussion section are not warranted.

The authors often compare their work to concept mapping. They say that the network measures allow to assess the quality of written explanations and point to analogous research on concept maps. However, e.g., Ruiz-Primo 1996 provides strong critiques of concept maps as assessment tools. Does this then also apply to the element maps? I don't think so but the authors should use this as an opportunity and tell us how element maps avoid problems of concept mapping.

Advising students to include just more relevant concepts (L 208) seems strange to me as it could lead to explanations that have a lo of terminology but little meaning. Isn't it more about the structure? 

Lastly, I think that Figure 1 provides a very good overview as to the distinction between concept maps and element maps.

Author Response

Reviewer (always italic): However, when reading the introduction and discussion I felt that key references where missing relative to which the authors should situate their work to make a better contribution. More specifically, I miss a discussion of the influential work of Toulmin on explanations and argumentation in general. In science education, the authors do not discuss the work of e.g. Kulgemeyer or McNeil who introduces a framework for teaching explanations, or Yao who proposes a learning progression for explanations.

Reply to the reviewer:

Thank you for this comment. According to Rocksen (2016), explanations can be understood in three ways: asinstructional practice, as scientific explanations or in everyday situations. Kulgemeyer (2013, 2018), for example, can be located in the first way, underpinning, that his own standpoint (within the SCC framework) has to be distinguished from scientific explanations. By contrast, we locate ourselves in the second understanding of scientific explanations. This is now clarified in the text.

There are several frameworks for explanations in science education, e.g. McNeil et. al (2006), Yao (2018), Gilbert (1998), or from the model perspective of e.g. Passmore (2014). Each of these frameworks can make a valuable contribution to the understanding of explanations and provide a basis for investigations. A commitment to one of them is neither necessary nor useful. Moreover, CER framework from McNeil takes an integrative approach to argumentation and explanation. Osborne (2011), by contrast, pointed out that the two should be separated. We follow this advice. In addition, the CER framework structures argumentations and explanations into claims, evidence and reasons. Each of these parts can consist of several statements, and each statement is again characterized by links between objects, properties and changes. And this is exactly what we make more accessible and quantitatively measurable. However, we take the reviewer's suggestion as an opportunity to emphasize this aspect in the discussion. Furthermore, we added some sentences into the introduction and outlook that show our awareness of these several frameworks.

Similarly, explanation maps by Clark are not mentioned.

Reply to the reviewer:

We couldn’t find something like explanation maps by Clark. We also asked other contributors of the special issue on networks if they know that, but they don’t. Also, in already published papers of this special issue, where explanation maps might be relevant, we couldn’t find it. Did you mean those argumentation patterns of DB Clark?

Since the paper (at least implicitly) makes the argument that the element maps provide new understanding I would like to see a discussion of what it is that we gain (as science education researchers AND for classroom practice) when we use element maps instead of e.g., the CER Framework (McNeil) to judge the quality of an explanation. Such a discussion would largely improve the papers chance of meaningfully contributing to the literature. Right now, a statement like "Our novel representation of written explanations provides unique and quantitative measures to the quality of their elementary structure. This makes key features of a written explanation accessible and comparable in a numerical, as well as visual, manner." does not convince me until I read an argument why e.g. the CER framework does not give me all the information about the quality of an explanation that I would want.

Reply to the reviewer:

This is a good point, which we like to clarify. We do not claim that it would be better to use element maps instead of the CER framework. And, we are not introducing a new framework, but a method to visualize linguistic products and make them more accessible. We don't see why and how this method conflicts with a particular framework. Furthermore, we do not want to replace other methods of assessing quality with the features of explanations that become visible through the network approach. Rather, it offers additional possibilities and provides additional information that would otherwise simply be difficult to access. As far as we can judge, the CER framework offers many helpful accesses to explanations, but it does not provide information about their structure at the elementary level, about the quantitative measure of complexity and the key function of certain elements, as element maps do. If the reviewer finds a publication that shows the opposite, we are happy to analyze and include it.

However, we have included a paragraph that emphasizes the potential compatibility and benefits of element maps as a method and different approaches as conceptual frameworks.

I wonder whether science education researchers are really good "experts" for scientific explanations? Why not ask physicists etc?

Reply to the reviewer:

We have also thought about this several times. We formulated an argument for our choice of the science education researches as experts and inserted it into the text. For the information of the reviewer: In a pilot study we also asked physics professors for explanations. However, these explanations were anything but satisfactory. According to the usual understanding, experts in a field are characterized by the fact that they have specialized knowledge in that field. To be honest, which physicist should have special knowledge of apparent depth? Quantum optics physicist? It is very plausible to assume that science education researchers and lecturers are indeed experts in explaining everyday phenomena.

There should be a more elaborate in justifying their network measures in the methods section, e.g., the authors say that high betweenness centrality indicates nodes that connect different parts of the networks but do not say why connecting different parts of the element maps may be important in the context of explanations.

Reply to the reviewer:

We agree to the need for clarification of network measures, especially the betweenness centrality. Since our method section only contains a description of the survey design and transformation procedure, we clarified this point in theory section (where network measures are defined), and in the discussions and conclusions. But, please consider that we do not define a high or low level of these measures as important a priori, but rather explore differences in these network characteristics between experts and look for plausible results from other network applications. We have already described it in detail for every measure.

Why focus on the explanations of the apparent depth phenomenon only if data on others was collected?

Reply to the reviewer:

Other data were collected based on explanations for similar, but different phenomena. Because of the usual variable control strategy in research contexts we decided to keep the phenomenon constant as one variable for the results presented here. Thus, including other phenomena is not the scope of this publication and would address another research question.

The authors state (L 145) that they do not judge the normative correctness of the propositions but refer to a dissertation. This is ok, but they should discuss the potential and limitations of doing so more elaborately.

Reply to the reviewer:

We have clearified this in both, the „Materials and Methods Section“ and the „Outlook“ (which is now called „Limitations and Outlook“. Please note, that we have done both, a content analysis including an evaluation of the propositions correctness. In this paper we have limited ourselves to the quantitative presentation of structural features only, otherwise the article would have become much longer and more difficult to follow for the reader.

Figure 5 uses box plots to represent the data from 3 experts. as box plots try to depict a distribution this is not warranted here. Instead, the authors should just plot the three data points.

Reply to the reviewer: done

In the discussion (L 223) the authors discuss how their network measures relate to structural knowledge. Basically, this is inconsistent with not judging the correctness of the propositions (L 145). How can we talk about knowledge if the maps, strictly following the methods section and not judging the correctness of propositions, could have been created from totally wrong explanations that are not even on the topic? I understand that the authors do want to focus on the structural aspects but somehow this inconsistency needs to be solved, as in the current state the interpretations in the discussion section are not warranted.

Reply to the reviewer:

It is absolutely correct that without an assessment of propositions’ correctness, no complete statement can finally be made as to whether the underlying explanation is complete nonsense. However, the statement that the reviewer criticizes here (L 223) concerns experts. To assume from the outset that experts provide a completely false explanation is not reasonable. Furthermore, in the results presented here we restrict ourselves to qualities of structure, not to those of content. Nevertheless, what we have stated transparently throughout the paper what also applies here: without an assessment of the quality of the content, this analysis remains incomplete. At this critical point, however, we have now included a small insight into the content analysis, which may refute the objection.

The authors often compare their work to concept mapping. They say that the network measures allow to assess the quality of written explanations and point to analogous research on concept maps. However, e.g., Ruiz-Primo 1996 provides strong critiques of concept maps as assessment tools. Does this then also apply to the element maps? I don't think so but the authors should use this as an opportunity and tell us how element maps avoid problems of concept mapping.

Reply to the reviewer:

Well, Ruiz-Primo did not criticize concept maps as assessment tools per se, but rather the missing estimation (or report) of the reliability and validity for CM’s as assessment tools in their varying appearances. We checked the validity and reliability of identifying (and further categorizing) elements and relations for the entire study of 64 explanations (for more explanations), which our presented sample is a part of. However, we pointed to that in the „Materials and Methods section“.

Advising students to include just more relevant concepts (L 208) seems strange to me as it could lead to explanations that have a lo of terminology but little meaning. Isn't it more about the structure?

Reply to the reviewer:

Yes, but size is a characteristic of the structure. And in the case of particularly small maps, it is plausible to look first to see whether many elements, i.e. those used by experts, are missing in the learner's explanation and, if so, need to be integrated. In other words, if you as a teacher or lecturer look at your own map, compare it to one made from a learners explanation and see, that this one has only 10 percent of the size of your map, you can directly conclude that presumably some elements are missing. And please consider, that this is not necessarily visible in the length of the original written explanation, since this could contain the same or redundant statements over and over and, hence, becomes a „long“ explanation in this way.

Reviewer 2 Report

I am concerned that you are ignoring the acuracy of the statements made in the explanations. One could write total nonsense statments using the same words over and over again and score very highly using your method. Parsimony will lead to lower scores, but on the other hand, parsimony is a valued characteristic of scientific explanations. You need to better justify why one can treat a relation as if it were an entity = a vertex. They are ontologically different. This is very different from what is done in concept maps, which you often refer to to justify your analyses. You should have used more than 3 experts. Three does not provide you with enough variance. You wrote: "By looking at the different characteristics of the element maps, teachers could give individual feedback to students to improve their written explanations." I fail to see how the analyses you did can, at this stage, be translated into concrete recommendations that teachers can give students. Also, for a teacher to do something like this, it needs to be relatively quick and simple. If this is the case, why did you use such a small sample for the study?

Author Response

Reviewer (always in italics): I am concerned that you are ignoring the accuracy of the statements made in the explanations. One could write total nonsense statements using the same words over and over again and score very highly using your method. Parsimony will lead to lower scores, but on the other hand, parsimony is a valued characteristic of scientific explanations.

Reply to the reviewer:

On this point we fully agree with the reviewer. Repeating the same statement several times should not be taken as a better explanation. This is already ensured by consolidating entities and relationships of equal meaning in the maps. We have added a sentence that emphasizes this characteristic of element maps more strongly.

You need to better justify why one can treat a relation as if it were an entity = a vertex. They are ontologically different. This is very different from what is done in concept maps, which you often refer to to justify your analyses.

Reply to the reviewer:

Please consider Figure 3, where the different ontological categories of entities and relationships are represented by different colors. The distinction does not necessarily have to be made by assigning them to vertices and edges. However, we have inserted a few clarifying sentences near this figure.

You should have used more than 3 experts. Three does not provide you with enough variance.

Reply to the reviewer:

We fully agree. That’s why we have made transparent that the sample size is a limitation of the study. We  emphasized the need of increasing the number of experts (and learners) again.

You wrote: "By looking at the different characteristics of the element maps, teachers could give individual feedback to students to improve their written explanations." I fail to see how the analyses you did can, at this stage, be translated into concrete recommendations that teachers can give students.

Reply to the reviewer:

We have refined the statement. However, please bear in mind that the development and effectiveness testing of suitable interventions can only be a next step and is not in the scope of this report.

Also, for a teacher to do something like this, it needs to be relatively quick and simple. If this is the case, why did you use such a small sample for the study?

Reply to the reviewer:

This is true. As we explain in the section Methods and Materials, this is part of a larger investigation (with 64 explanations) that examines explanations of various related optical phenomena. However, comparing different explanations would have gone beyond the scope of this paper. Furthermore, we only make the statement that transforming explanations into element maps is simple for small explanations by hand. Many explanations in school might often consist of just a few sentences. In the sample, however, there are also longer explanations (at least, the experts’). For better implementability, e.g. for teachers, we plan to develop a software-based method for creating element maps. We point this out in Outlook.

Reviewer 3 Report

I really enjoyed reading this paper. I think the context and significance are described clearly and well, and the appropriate literature drawn upon. The approach used makes an original contribution to knowledge.

In a sense the findings are unsurprising: explanations given by experts are not just more correct than those given by novices, they are more elaborate and detail and have a higher level of interconnectedness between elements. Being able to explore these differences and describe them more clearly and in more detail, however, is valuable.

There are a few minor issues with writing and English expression: the English is very good, but occasionally ungrammatical or not standard. Terms like 'intertwinement' and 'betweenness centrality' could perhaps be replaced by more accessible terms. Alternatively, they can be clearly defined on their first usage. 

In Figure 3, 'liftet' is used instead of 'lifted'.

The paragraph on Lines 29-34 has some repetition. 

The figures are very clear and communicative.

The results are clearly analysed and their signficance explained, and the implications for both practice and future research clearly outlined.

Author Response

Reviewer: There are a few minor issues with writing and English expression: the English is very good, but occasionally ungrammatical or not standard. Terms like 'intertwinement' and 'betweenness centrality' could perhaps be replaced by more accessible terms. Alternatively, they can be clearly defined on their first usage.  

Reply to the reviewer:

Thanks for this comment. Indeed, we spend some time thinking about how to describe the complexity without using an already established term from network theory with another meaning (like e.g.: density, complexity). Finally intertwinement and entanglement where left. Here we decided for intertwinement, because it captures the concept a bit better. By contrast, betweenness centrality is a usual term (one of many centrality measures) in network theory and it is described in our list of network measures. However, for clarification of the term intertwinement, we add a sentence in the list of the network measures.

Reviewer: The paragraph on Lines 29-34 has some repetition. / In Figure 3, 'liftet' is used instead of 'lifted'.

You’re right. We’ve improved that.

Round 2

Reviewer 1 Report

Dear authors, 

I appreciate how you addressed my comments and issues. Before I refer to some minor points that remain I wanted to let you know that, yes I was referring to the argumentation patters of DB Clark. He calls them explanation maps (480) and also element maps (p 482). So there might be a better name for your maps that could help to avoid confusion. 

After reading your answers to my comments, I was struck by the title of the paper. After all, are you really measuring the quality of the explanation - wouldn't absolute quality have to consider structure AND content? Thus, for me, to measure the quality you would either need to include a content analysis or demonstrate how the network measures compare to another measure of explanation quality. Unless you do this, I strongly suggest something like "Measuring the characteristics of explanations with elements maps" as a more appropriate title. 

Lastly, I think combining the content and network analysis would make for a much stronger paper and I would encourage you to pursue that. 

Author Response

We changed the title in the document and (in a footnote) clarified that our element maps should not be confused with those of Clark, which are no networks and thus do not conflict our usage of the term.